# Phosphoproteomics Profile of Chicken Cecum in the Response to *Salmonella enterica* Serovar Enteritidis Inoculation

**DOI:** 10.3390/ani13010078

**Published:** 2022-12-25

**Authors:** Xiuxiu Miao, Ya’nan Zhao, Huilong Li, Yanru Ren, Geng Hu, Jingchao Yang, Liying Liu, Xianyao Li

**Affiliations:** 1College of Animal Science and Technology, Shandong Provincial Key Laboratory of Animal Biotechnology and Disease Control and Prevention, Shandong Agricultural University, Tai’an 271018, China; 2Shandong Animal Husbandry General Station, Jinan 250010, China; 3College of Life Sciences, Shandong Agricultural University, Tai’an 271018, China; 4Key Laboratory of Efficient Utilization of Non-Grain Feed Resources (Co-Construction by Ministry and Province), Ministry of Agriculture and Rural Affairs, Shandong Agricultural University, Tai’an 271018, China

**Keywords:** chicken, *Salmonella enterica* serovar Enteritidis, cecum, phosphoproteomics

## Abstract

**Simple Summary:**

*Salmonella enterica* serovar Enteritidis (*S*. Enteritidis) is one of the most common pathogens associated with poultry health and foodborne salmonellosis worldwide. It can cause a major public health problem and economic losses in the world. Traditional methods such as vaccines and antibiotics can make poultry and livestock resistant to drugs. Improving the genetic resistance of livestock and poultry is an effective supplement to traditional control. We found a variety of inflammatory genes were triggered at the mRNA level infected with *S*. Enteritidis of chickens previously. However, the phosphoproteomics profiles for chickens infected with were not clear. Phosphoproteomics was employed to dissect the molecular regulation mechanisms of F1 cross chicken cecum infected with *S*. Enteritidis. Functional analysis and functional classification were performed for those differentially phosphorylated proteins (DPPs). The findings in this study provide a crucial theoretical foundation to understand the molecular mechanism and epigenetic regulation in response to *S*. Enteritidis inoculation in chicken.

**Abstract:**

*Salmonella enterica* serovar Enteritidis (*S*. Enteritidis) is a foodborne pathogen, which can cause great threats to human health through the consumption of contaminated poultry products. This research combines TMT labeling, HPLC and mass-spectrometry-based phosphoproteomics on cecum of the F1 cross of Guangxi Yao chicken and Jining Bairi chicken. The treated group was inoculated with 0.3 mL inoculum *S*. Enteritidis, and the control group was inoculated with 0.3 mL phosphate-buffered saline (PBS). A total of 338 differentially phosphorylated modification sites in 243 differentially phosphorylated proteins (DPPs) were chosen for downstream analyses. A total of 213 sites in 146 DPPs were up-regulated and 125 sites in 97 DPPs were down-regulated. Functional analysis was performed for DPPs based on gene ontology (GO), Kyoto Encyclopedia of Genes and Genomes (KEGG) pathways, and the protein domain. The DPPs were mainly enriched in immune- and metabolic-related GO-BP (biological process) and KEGG pathways. We predicted and classified the subcellular structure and COG/KOG of DPPs. Furthermore, protein–protein interaction network analyses were performed by using multiple algorithms. We identified 71 motifs of the phosphorylated modification sites and selected 18 sites randomly to detect the expression level through parallel reaction monitoring (PRM). S. Enteritidis inoculation caused phosphorylation alteration in immune- and metabolic-related proteins. The invasion of S. Enteritidis may be actualized by inducing cecum cell apoptosis through the endoplasmic reticulum pathway, and chickens could resist the invasion of S. Enteritidis by affecting the function of ECM receptors. The findings herein provide a crucial theoretical foundation to understand the molecular mechanism and epigenetic regulation in response to *S*. Enteritidis inoculation in chickens.

## 1. Introduction

*Salmonella enterica* serovar Enteritidis (*S*. Enteritidis) is one of the most common pathogens associated with poultry health and foodborne salmonellosis worldwide [1], which can cause a major public health problem in the world. Foodborne zoonotic diseases can be transferred into the food chain at the stage of poultry farming [2]. Poultry is a core reservoir of *S*. Enteritidis and *S*. Enteritidis is linked to approximately seventy percent of salmonellosis foodborne cases [3]. In addition, *S*. Enteritidis infection in chicks can result in asymptomatic cecal colonization, gut dysbiosis, intestinal mild inflammation [4,5], and gut barrier damage along with the invasion of internal organs and poultry carcasses [6]. The frequent utilization of antimicrobials in agriculture for growth promotion and bacterial infection treatment can lead to the evolution of the susceptible to antimicrobial-resistant pathogens [7]. Therefore, improving the genetic resistance of livestock and poultry is an effective supplement to traditional control, and it is important for the selection and breeding of disease-resistant strains of livestock and poultry.

We found that a variety of inflammatory genes were triggered at the mRNA level infected with *S.* Enteritidis of chickens in our previous research [8]. Additionally, the isomiRs of gga-miR-146b-5p might act to sustain the RIG-I-like receptor signaling and type I interferon induction by repressing the *USP3* transcript [9]. Although the sequencing results at the RNA level have been described, the phosphoproteomics profiles of chickens infected with S. Enteritidis were not clear. The recent development of mass-spectrometry-based proteomic techniques has allowed us to characterize the phosphoproteomics of specific cells or tissues and identify protein phosphorylation in a high-throughput way [10]. Protein phosphorylation is a critical and reversible post-translational modification (PTM) involved in most dynamic regulation of biological processes, such as cell division, cell growth, apoptosis, signal transduction, response to extracellular stimulation, and altering enzymatic activity [11,12]. It has been reported that dehydroepiandrosterone affected both protein synthesis and phosphorylation level to decrease fat deposition in broiler chickens during embryonic development [13]. Several phosphoproteomics studies have been performed on duck eggs, the phosphoproteins in duck egg white were mainly involved in the binding, transport activity, biological regulation, and metabolic processes [14], and the phosphoproteins in duck egg yolk were mainly involved in binding, catalytic, immune response, and metabolic activity [15]. In our study, phosphoproteomics was employed to dissect the molecular regulation mechanisms of F1 cross chicken cecum infected with *S*. Enteritidis.

## 2. Materials and Methods

### 2.1. Animal Inoculation and Sample Collection

An F1 cross (Guangxi Yao ♀ × Jining Bairi ♂) was used in the current study, which was provided by Jining Bairi Chicken Breeding Co., Ltd. (Jining, China). A total of 90 *S*. Enteritidis-negative 2-day-old chickens were randomly divided into two groups and raised in two separate incubators with the same environmental conditions and free access to sterilized feed and water. There were a total of 38 chickens in the control group and 52 chickens in the treated group. The *S.* Enteritidis (CVCC3377) used for inoculation was provided by the China Veterinary Culture Collection Center (http://cvcc.ivdc.org.cn/, accessed on 10 October 2018). Each chicken in the treated group was inoculated with 0.3 mL inoculum of 6.37 × 10^7^ colony-forming units (cfu) /mL *S*. Enteritidis solution, and each chicken in the control group was inoculated with 0.3 mL sterile phosphate-buffered saline (PBS). Three chickens in the treated group and three chickens in the control group were euthanized by cervical dislocation on 3 days post-inoculation; the cecum was collected from each bird, snap-frozen, and kept in −80 °C. All animal experiments were approved by the Laboratory Animal Management and Use Committee of Shandong Agricultural University (Permit Number: SDAUA-2018-058).

### 2.2. Protein Extraction and Trypsin Digestion

The sample was ground using liquid nitrogen into cell powder and then transferred to a 5 mL centrifuge tube. After that, four volumes of lysis buffer (8 M urea (Sigma Chemical CO, St. Louis, MO, USA), 1% Protease Inhibitor Cocktail (Calbiochem, Sigma-Aldrich, St. Louis, MO, USA), 3 μM TSA, 50 mM NAM, and 2 mM EDTA) was added to the cell powder, followed by sonication three times on ice using a high-intensity ultrasonic processor (Scientz, Ningbo, China). The remaining debris was removed through centrifugation at 12,000× *g* at 4 °C for 10 min. Finally, the supernatant was collected and the protein concentration was determined with a BCA kit (Beyotime, Nanjing, China) according to the manufacturer’s instructions.

For digestion, the protein solution was reduced with 5 mM dithiothreitol (Sigma Chemical CO, St. Louis, MO, USA) for 30 min at 56 °C and alkylated with 11 mM iodoacetamide (Sigma Chemical CO, St. Louis, MO, USA) for 15 min at room temperature in darkness. The protein sample was then diluted by adding 100 mM TEAB (Sigma Chemical CO, St. Louis, MO, USA) to urea concentration less than 2 M. Finally, trypsin (Yaxin Bio, Shanghai, China) was added at a 1:50 trypsin-to-protein mass ratio for the first digestion overnight and a 1:100 trypsin-to-protein mass ratio for the second 4 h of digestion.

### 2.3. TMT Labeling and High-Performance Liquid Chromatography (HPLC) Fractionation

After trypsin digestion, the peptide was desalted using a Strata X C18 SPE column (Phenomenex, Torrance, CA, USA) and vacuum-dried. The peptide was reconstituted in 0.5 M TEAB and processed according to the manufacturer’s protocol for the TMT kit (Thermo Fisher Scientific, Waltham, MA, USA). Briefly, one unit of TMT reagent was thawed and reconstituted in acetonitrile (Thermo Fisher Scientific, Waltham, MA, USA). The peptide mixtures were then incubated for 2 h at room temperature and pooled, desalted, and dried through vacuum centrifugation.

The tryptic peptides were fractionated into fractions by high-pH reverse-phase HPLC using a Thermo Betasil C18 column (5 μm particles, 4.6 mm ID, and 250 mm length). Briefly, peptides were first separated with a gradient of 8% to 32% acetonitrile (pH 9.0) over 60 min into 60 fractions. Then, the peptides were combined into 6 fractions and dried by vacuum centrifuging.

### 2.4. Affinity Enrichment of the Phosphopeptides

The peptide mixtures were first incubated with IMAC microsphere suspension with vibration in loading buffer (50% acetonitrile/6% trifluoroacetic acid (Sigma-Aldrich, St. Louis, MO, USA)). The IMAC microspheres with enriched phosphopeptides were collected using centrifugation, and the supernatant was removed. To remove nonspecifically adsorbed peptides, the IMAC microspheres were washed with 50% acetonitrile/6% trifluoroacetic acid and 30% acetonitrile/0.1% trifluoroacetic acid, sequentially, three times. To elute the enriched phosphopeptides from the IMAC microspheres, elution buffer containing 10% NH_4_OH was added and the enriched phosphopeptides were eluted with vibration. The supernatant containing phosphopeptides was collected and lyophilized. We demineralized according to the C18 ZipTips (Millipore, MA, USA) instructions and drained, and lyophilized for LC-MS/MS analysis.

### 2.5. LC-MS/MS Analysis

The tryptic peptides were dissolved in solvent A (0.1% formic acid (Honeywell Fluka, Seelze, Germany) in 2% acetonitrile) and directly loaded onto a homemade reversed-phase analytical column (15 cm length, 75 μm i.d.). The gradient was comprised of an increase from 4% to 22% solvent B (0.1% formic acid in 90% acetonitrile) over 38 min, 22% to 32% in 14 min, and climbing to 80% in 4 min, then holding at 80% for the last 4 min, all at a constant flow rate of 450 nL/min on an EASY-nLC 1200 UPLC (ultra-performance liquid chromatography) system.

The peptides were subjected to an NSI source followed by tandem mass spectrometry (MS/MS) in Q ExactiveTM Plus (Thermo Fisher Scientific, Waltham, MA, USA) coupled online with the UPLC. The electrospray voltage applied was 2.0 kV. The *m*/*z* scan range was 350 to 1600 for a full scan, and intact peptides were detected in the Orbitrap at a resolution of 60,000. The fixed starting point of the scan range of the secondary mass spectrometry was 100 *m*/*z*, and the resolution was set to 30,000. Peptides were then selected for MS/MS using the normalized collision energy (NCE) setting as 28 and the fragments were detected in the Orbitrap at a resolution of 30,000. A data-dependent procedure, data-dependent acquisition (DDA), was carried out which alternated between one MS scan followed by 20 MS/MS scans with 15.0 s dynamic exclusion. Automatic gain control (AGC) was set at 1E5, and the signal threshold was set to 50,000 ions/s.

### 2.6. Database Search

The resulting MS/MS data were processed using Maxquant search engine (v.1.5.2.8). Tandem mass spectra were searched against the chicken database (Gallus_gallus_PR, 29475 sequences) concatenated with a reverse decoy database. Trypsin/P was specified as cleavage enzyme allowing up to 2 missing cleavages. The minimum length of the peptide was set as 7 amino acid residues, and the maximum modification number of the peptide segment was set to 5. The mass tolerance for precursor ions was set as 20 ppm in the first search and 5 ppm in main search, and the mass tolerance for fragment ions was set as 0.02 Da. Carbamidomethyl on Cys was specified as fixed modification and N-terminal acetylation modification, oxidation, and deamidation (NQ) on Met were specified as variable modifications. FDR was adjusted to <1% and the minimum score for modified peptides was set > 40. Proteins with a fold change > 1.50 and *P* adj < 0.05 were considered to exhibit a significant difference and subjected to the subsequent bioinformatics analyses.

### 2.7. Functional Classification of Differentially Phosphorylated Proteins (DPPs) Corresponding to Differentially Phosphorylated Modification Sites

Wolfpsort v.0.2 (http://www.genscript.com/psort/wolf_psort.html, accessed on 15 October 2022) and CELLO v.2.5 (http://cello.life.nctu.edu.tw/, accessed on 15 October 2022)were used to predict and classify the subcellular structure of DPPs. COG refers to clusters of indigenous groups of proteins. COG is divided into two types, one is prokaryotic and the other is eukaryotic. Prokaryotes are generally referred to as COG databases, eukaryotes are commonly referred to as KOG databases. We compared and analyzed the databases to classify the COG/KOG functions of the DPPs.

### 2.8. Functional Enrichment Based on Gene Ontology (GO), Kyoto Encyclopedia of Genes and Genomes (KEGG) Pathway, and Protein Domain of DPPs

Proteins were classified by GO annotation into three categories: biological process, cellular component, and molecular function. For each category, a two-tailed Fisher’s exact test was employed to test the enrichment of the differentially modified protein against all identified proteins. The GO term with a corrected *P* adj < 0.05 was considered significant.

The KEGG database [16,17,18] was used to identify enriched pathways with a two-tailed Fisher’s exact test to test the enrichment of the differentially modified protein against all identified proteins. The pathway with a corrected *P* adj < 0.05 was considered significant. These pathways were classified into hierarchical categories according to the KEGG website.

For each category of proteins, the InterPro (a resource that provides functional analysis of protein sequences by classifying them into families and predicting the presence of domains and important sites) database was searched and a two-tailed Fisher’s exact test was employed to test the enrichment of the differentially modified protein against all identified proteins. Protein domains with a corrected *P* adj < 0.05 were considered significant.

### 2.9. Protein–Protein Interaction (PPI) Network

All DPPs and modified protein database accessions or sequences were searched against the STRING database version 10.5 for PPIs analysis. Only interactions between the proteins belonging to the searched data set were selected, thereby excluding external candidates. STRING defines a metric called “confidence score” to define interaction confidence,;we fetched all interactions that had a confidence score ≥ 0.4. The interaction network form STRING was visualized using Cytoscape 3.7.2. The MCODE algorithm was used to analyze the characteristics of the networks and find densely connected regions. The key biological functions that each module participated in were labelled.

### 2.10. Motif Analysis

Soft MoMo (motif-x algorithm) was used to analyze the model of sequences comprising of amino acids in specific positions of modify-21-mers (10 amino acids upstream and downstream of the site, but phosphorylation with modify-13-mers with 6 amino acids upstream and downstream of the site) in all protein sequences. Additionally, all the database protein sequences were used as a background database parameter. The minimum number of occurrences was set to 20 and *p* < 10^−6^. Emulate original motif-x was ticked, and other parameters with default.

## 3. Results

### 3.1. Identification of Phosphorylated Modification Sites and DPPs of Cecum in Chickens

The LC-MS/MS data analysis generated a total of 25,709 matched spectra, 10,334 peptides and 8438 modified peptides. We identified 7329 phosphorylated modification sites, corresponding to 3155 proteins. There were 338 differentially phosphorylated modification sites (fold change > 1.50 and *P* adj < 0.05) in 243 DPPs which were chosen for downstream analysis. Among the quantified proteins, 213 sites in 146 DPPs were up-regulated and 125 sites in 97 DPPs were down-regulated (Figure 1).

### 3.2. Subcellular Structure Localization and COG/KOG Functional Classification of DPPs

In the phosphoproteomics results, more than half of the up-regulated DPPs were localized in the nucleus (65/146) and cytoplasm (37/146). Additionally, there were 13 up-regulated DPPs localized in the plasma membrane and extracellular, respectively (Figure 2A). Most of the down-regulated DPPs localized in the nucleus (29/97), cytoplasm (27/97), and plasma membrane (24/97). There were seven, six, and four down-regulated DPPs localized in the cytoplasm, nucleus, mitochondria, and extracellular, respectively (Figure 2B).

COG/KOG functional classification was divided into four categories: information storage and processing, cellular processes and signaling, metabolism, and poorly characterized. There were 54 up-regulated DPPs assigned to COG/KOG functional categories, five up-regulated DPPs were assigned to metabolism-related classifications (Figure 3A). There were 39 down-regulated DPPs assigned to COG/KOG functional classifications. Similar to the up-regulated DPPs, four down-regulated DPPs were assigned to metabolism-related classifications (Figure 3B).

### 3.3. Functional Analysis of DPPs in Chicken Cecum

In order to understand the possible roles of DPPs in chickens post-infected with *S*. Enteritidis, proteins were annotated using the GO and the KEGG pathway database to further investigate their functions. The up-regulated DPPs were enriched in 14 GO-BP (biological process) terms (*P* adj < 0.05) (Figure 4A). The enriched GO-BP terms could be divided into four groups: defense responses, activation of immune cells, antioxidant-related responses, and others. The GO-BP terms related to defense responses included defense response to bacterium, defense response to other organisms, and response to bacterium. The GO-BP terms related to antioxidants were cellular response to oxygen radical, response to oxygen radical, response to superoxide, and regulation of response to oxidative stress. Myeloid leukocyte activation and mast cell activation could reflect the immune response after *S*. Enteritidis infection. We identified three GO-BP terms from down-regulated DPPs (*P* adj < 0.05) (Figure 4B), adherens junction organization, retrograde vesicle-mediated transport, Golgi to ER, and positive regulation of GTPase activity. The enriched KEGG pathways (*p* < 0.05, *P* adj > 0.05) of up-regulated DPPs were related to metabolism, such as Arginine and proline metabolism, retinol metabolism, glycolysis/gluconeogenesis, and arachidonic acid metabolism. ECM–receptor interaction, the PPAR signaling pathway, the MAPK signaling pathway, and Salmonella infection were also enriched by up-regulated DPPs (Figure 4C). The down-regulated DPPs were enriched in the ABC transporters KEGG pathway (*P* adj < 0.05). Other enriched KEGG pathways (*p* < 0.05, *P* adj > 0.05) associated with down-regulated DPPs included fatty acid degradation, purine metabolism, pyrimidine metabolism, drug metabolism–other enzymes, the MAPK signaling pathway, the NOD-like receptor signaling pathway, and the Wnt signaling pathway (Figure 4D).

### 3.4. Protein Domain Analysis

The up-regulated DPPs were enriched in 12 protein domains (*P* adj < 0.05), involved HSP20-like chaperone, heat shock protein 70 kD, peptide-binding domain, and HSP20-like chaperone, which was related to the heat shock protein. N-CoR, the GPS2-interacting domain, EBP50, C-terminal, and Thioredoxin-like fold were considered as significant protein domains (Figure 5A). The number of enriched protein domains associated with down-regulated DPPs was more than that of the up-regulated DPPs. The down-regulated DPPs were enriched in 28 protein domains, involved ABC transporter type 1, the transmembrane domain, ABC-transporter-like, Vacuolar-protein -sorting-associated protein 13, C-terminal, Vacuolar-protein-sorting-associated protein 13, N-terminal domain, Vacuolar-protein-sorting-associated protein 13, second N-terminal domain, Zinc finger, UBR-type, Myosin head, and motor domain (Figure 5B).

### 3.5. Protein–Protein Interactions (PPIs) Networks of the DPPs

To further understand the biological function of differentially regulated phosphoproteins in chickens infected with *S*. Enteritidis, We assembled the PPI networks of the identified DPPs (Figure 6). Three major connected subnetworks, including protein processing in the endoplasmic reticulum, tight junction, and endocytosis of differentially changed phosphoproteins, were enriched and visualized using Cytoscape software. Up-regulated phosphoproteins HSPA8, HSPB1, HSPH1, GAPDH, SOD1, PRDX1, and down-regulated phosphoprotein HTT were involved in protein processing in the endoplasmic reticulum subnetwork. The up-regulated phosphoproteins TJP2, DNAJC5, RAD23A, MAP1S, SLC4A1, SLC9A1, SLC9A3R1, HBAA, EIF4G1, CDK18, FAS, THRAP3, CHGA, AHSG, and KNG1, and the down-regulated phosphoproteins PPP2R5A, HECTD4, NCOR2, MYO1D, were involved in the networks.

### 3.6. Motif Analysis of the Phosphorylated Modification Sites

We took advantage of the large number of phosphorylated modification sites identified in this work to carry out a bioinformatics analysis to perform the motif-type analysis. We identified 64 conserved motifs based on 7181 phospho-serine (pS)-phosphorylated modification sites, and 7 conserved motifs were identified based on 527 phospho-threonine (pT)-phosphorylated modification sites (Appendix A), including motif No.1–64 from phospho-serine and motif No.65–71 from phospho-threonine. There were 23 pS motifs and 2 pT motifs were conserved (motif score ≥ 30.00). In particular, some motifs, the motif score > 32, and fold increase > 12 were strikingly conserved, which included [xxxxPx_S_PxKxxx] (motif No.1), [xxxRxx_S_PxPxxx] (motif No.2), [xxxGRx_S_Pxxxxx] (motif No.12), [xxxxxx_S_DEExxx] (motif No.14), [xxxxxx_S_xEDLxx] (motif No.17), [xxxxxx_S_EEExxx] (motif No.23), and [xxxxxx_S_EDExxx] (motif No.27).

Heat maps were generated to show the enrichment or depletion of specific amino acids neighboring the pS- and pT-phosphorylated modification sites. The amino acids aspartate (D), glutamate (E), lysine (K), proline (P), and arginine (R) had a tendency to be present in the proximity of the pS-phosphorylated modification sites (Figure 7A). The amino acids glutamate (E), lysine (K), proline (P), and arginine (R) were greatly represented in the areas proximal to the pT-phosphorylated modification sites. Lysine (K) and arginine (R) were greatly represented in the sequence surrounding the phosphorylated modification site but was greatly depleted in the -1 position. Cysteine (C), phenylalanine (F), isoleucine (I), leucine (L), methionine (M), valine (V), tryptophan (W), and tyrosine (Y) were notably depleted in the sequence surrounding the phosphorylation modification sites (Figure 7B).

### 3.7. Validation of the Phosphorylated Modification Sites with Parallel Reaction Monitoring (PRM)

To confirm the TMT phosphoproteomics profile results, we randomly selected 18 differentially phosphorylated modification sites from the phosphoproteomics using PRM (Figure 8). The phosphorylation of TJP2 at serine 382 and the phosphorylation of DNAJC5 at serine 10 and serine 15 were significantly up-regulated, and the phosphorylation of PBRM1 at serine 930 was significantly down-regulated (*p* < 0.05), which was consistent with the phosphoproteomics results. Although the fold change in these 14 differentially phosphorylated sites (*p* < 0.05) was less than 1.5, the expression trend was consistent with the sequencing results. The phosphorylation of CARHSP1 at serine 57, the phosphorylation of CASK at serine 571, the phosphorylation of KHSRP at serine 261, the phosphorylation of PDPK1 at serine 364, the phosphorylation of RAD23A at serine 186, the phosphorylation of RPM5 at serine 720, the phosphorylation of TMF1 at serine 484, the phosphorylation of MAP1S at serine 652, and the phosphorylation of DNAJC5 at serine 12 were significantly up-regulated; the phosphorylation of KPNA3 at serine 58, the phosphorylation of RAI14 at serine 391, the phosphorylation of FANCM at serine 996, the phosphorylation of PPP2R5A at serine 39, and the phosphorylation of HECTD4 at threonine 2504 were significantly down-regulated.

## 4. Discussion

Phosphoproteomic analysis is a powerful technique for investigating phosphoprotein expression patterns, and has been used for chickens in several studies [13,19,20]. The regulation mechanism of chicken cecum infected with *S*. Enteritidis is still being investigated; however, post-translational modifications, especially phosphorylation, might play an important role. Additionally, there has been little research to reveal the regulation mechanism of chicken cecum infected with *S*. Enteritidis in the phosphorylation levels of proteins. There has been a large amount of research on phosphorylated proteomics in human disease [21,22,23,24], bacteria [25], geese [26], goats [27,28], cattle [29], rats, and mice [30,31,32,33]. There were 62 phosphosites on 43 phosphoproteins which were determined to be up-regulated while 113 on 80 phosphoproteins were down-regulated in the research by Huang et al. [34]; the up-regulated phosphoproteins were fewer than the down-regulated phosphoproteins. Our results showed that the number of up-regulated phosphoproteins was higher than the down-regulated phosphoproteins, which was similar to Shi’s result in which 318 sites in 232 proteins were up-regulated and 204 sites in 159 proteins were down-regulated [35].

The enriched KEGG pathways of up-regulated DPPs were related to metabolism and immune response, and involved retinol metabolism, glycolysis/gluconeogenesis, ECM–receptor interaction, the PPAR signaling pathway, and the MAPK signaling pathway. The up-regulated DPPs CARHSP1, CASK, KHSRP, RAD23A, RBM5, and TMF1, the down-regulated DPPs FANCM, HECTD4, and PPP2R5A were closely connected to metabolism. Retinol, a fat-soluble vitamin, is converted to retinaldehyde and retinoic acid in the liver to maintain the normal physiological functions, such as growth, vision, immunity, and antioxidation [36,37]. Retinol and its derivatives can improve the capabilities of antioxidation and scavenging free radicals to reduce the risk of inflammation and oxidative stress [38]. The extracellular matrix (ECM) is a part of the connective tissue layers surrounding muscle fibers. The ECM provides support for the survival and activity of cells, and its effects include signal transduction that affects cell shape, function, metabolism, migration, proliferation, and differentiation [39]. The up-regulated DPPs were enriched in ECM–receptor interaction, which indicated that chickens could resist the invasion of *S*. Enteritidis by affecting the function of ECM receptors. PPAR signaling is closely related to fatty acid metabolism and biosynthesis; it maintains energy balance, insulin sensitivity, and the regulation of fatty acid biosynthesis [40]. It has been reported that the PPAR signaling pathway activated effective immune responses through MAPK signaling pathway to resist *Salmonella Typhimurium* infection in chickens [41], and the MAPK signaling pathway plays a vital role in the defense against various pathogens, including *S*. Enteritidis. MAPK signaling, essential for B-cell development and T-cell survival, was activated in resistant chickens after pathogenic infection [42]. It has been reported that the complicated interaction between the immune system and metabolism contributes to the immune responses to *S*. Enteritidis inoculation of chickens at 14 dpi at the onset of lay [8]. The enriched pathways may indicate that chickens resisted *S*. Enteritidis infection through regulating the expression of proteins via immune- and metabolic-related pathways.

The down-regulated DPPs were enriched in ABC transporters, the Wnt signaling pathway, the NOD-like receptor signaling pathway, etc. ATP-binding cassette (ABC) transporters constitute one of the largest and most ancient protein superfamilies found in all living organisms. They function as molecular machines by coupling ATP binding, hydrolysis, and phosphate release to the translocation of diverse substrates across membranes [43]. ABC transporters are membrane-spanning proteins involved in cholesterol homeostasis, the transport of various molecules in and out of cells and organelles, oxidative stress, and immune recognition [44]. The down-regulated DPPs ABCA3, ABCB1, and ABCC3 were involved in ABC transporters. We have reason to believe that the function of ABCA3, ABCB1, and ABCC3 will expend a great amount of energy to resist *S*. Enteritidis invasion after chickens are infected with *S*. Enteritidis, and it can provide energy and accelerate the transport of various substances. Wnt/β-catenin is a crucial signaling pathway in the growth of the body, cell differentiation, communication, and proliferation [45] and it was significantly altered in chicken cecum at day 4 after *S*. Enteritidis infection [46]. It is associated and involved in many diseases, including cancer, neurological, autoimmune, and inflammation diseases [47]. It has been suggested that the Wnt/β-catenin signaling pathway affects different viruses’ replication through different mechanisms [45]. The NOD-like receptor (NLR) family of proteins is a group of pattern recognition receptors (PRRs) known to mediate the initial innate immune response to cellular injury and stress [48,49]. The enriched Wnt signaling pathway and NOD-like receptor signaling pathway may indicate that *S*. Enteritidis inoculation would affect the chicken immune and metabolism through altering protein phosphorylation in the related signaling pathways.

Apoptosis (programmed cell death) is a universal form of death regulated by genes, which is crucial for normal organismal development and homeostasis [50]. Research has indicated that the hepatotoxicity of diclofenac sodium (DFS) may be achieved by inducing hepatocyte apoptosis through the endoplasmic reticulum pathway [38]. In our work, some of the up- and down-regulated DPPs were enriched in apoptosis and protein processing in endoplasmic reticulum KEGG pathways. The results suggested that the invasion of *S.* Enteritidis may be actualized by inducing cecum cell apoptosis through the endoplasmic reticulum pathway.

## 5. Conclusions

The phosphoproteomics profile in the response to *S*. Enteritidis inoculation in chickens was analyzed. S. Enteritidis inoculation promoted protein phosphorylation in the chicken cecum and caused phosphorylation alteration in immune- and metabolic-related proteins. The enriched Wnt signaling pathway and NOD-like receptor signaling pathway indicated that S. Enteritidis inoculation would affect the chicken immune response and metabolism through altering protein phosphorylation in the related signaling pathways. The invasion of S. Enteritidis may be actualized by inducing cecum cell apoptosis through the endoplasmic reticulum pathway, and chickens could resist the invasion of S. Enteritidis by affecting the function of ECM receptors. The findings herein provide a crucial theoretical foundation to understand the molecular mechanism and epigenetic regulation in response to *S*. Enteritidis inoculation in chickens.

## Figures and Tables

**Figure 1 animals-13-00078-f001:**
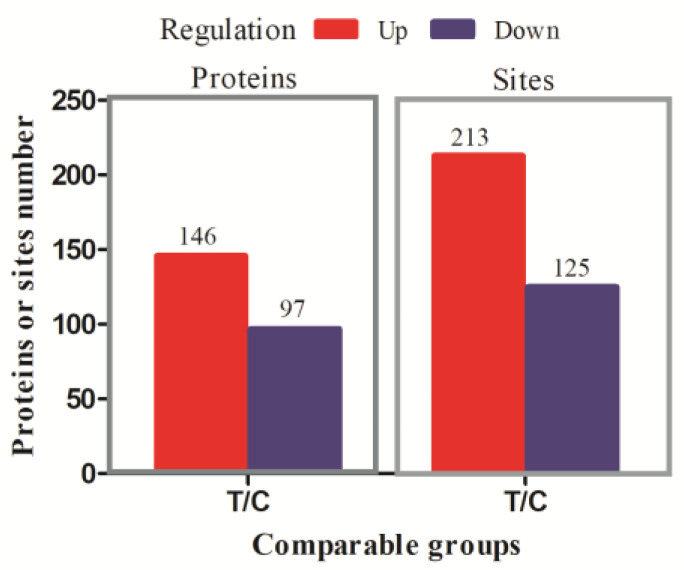
The results of differentially phosphorylated modification sites and differentially phosphorylated proteins (DPPs). A total of 213 sites in 146 DPPs were up-regulated and 125 sites in 97 DPPs were down-regulated (fold change > 1.50 and *P* adj < 0.05).

**Figure 2 animals-13-00078-f002:**
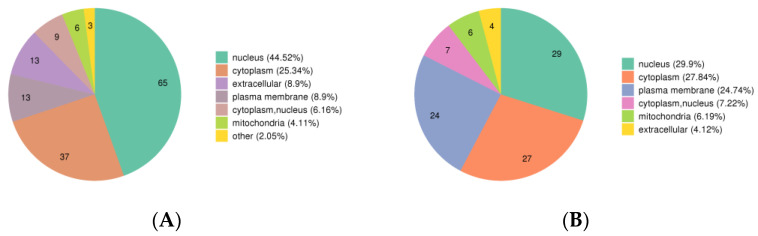
The subcellular structure localization classification of DPPs. (**A**) up-regulated DPPs. (**B**) down-regulated DPPs. Different colors indicate different subcellular structures. The number in the pie chart indicates the number of DPPs located in the subcellular structure.

**Figure 3 animals-13-00078-f003:**
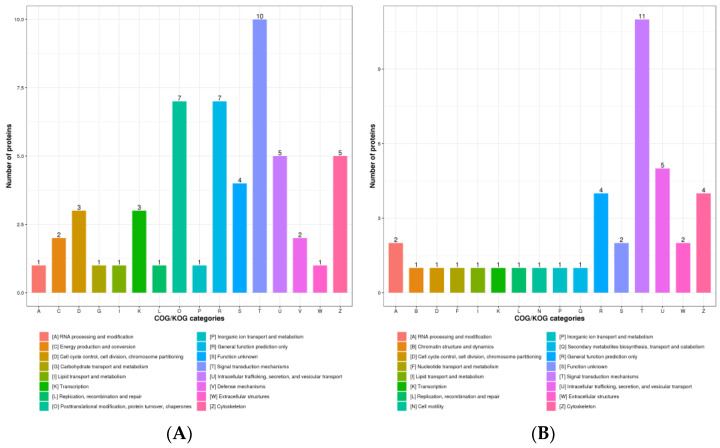
The COG/KOG classification of DPPs. (**A**) The COG/KOG classification of up-regulated DPPs. (**B**) The COG/KOG classification of down-regulated DPPs. The x-axis is the COG/KOG categories and the y-axis is the number of DPPs.

**Figure 4 animals-13-00078-f004:**
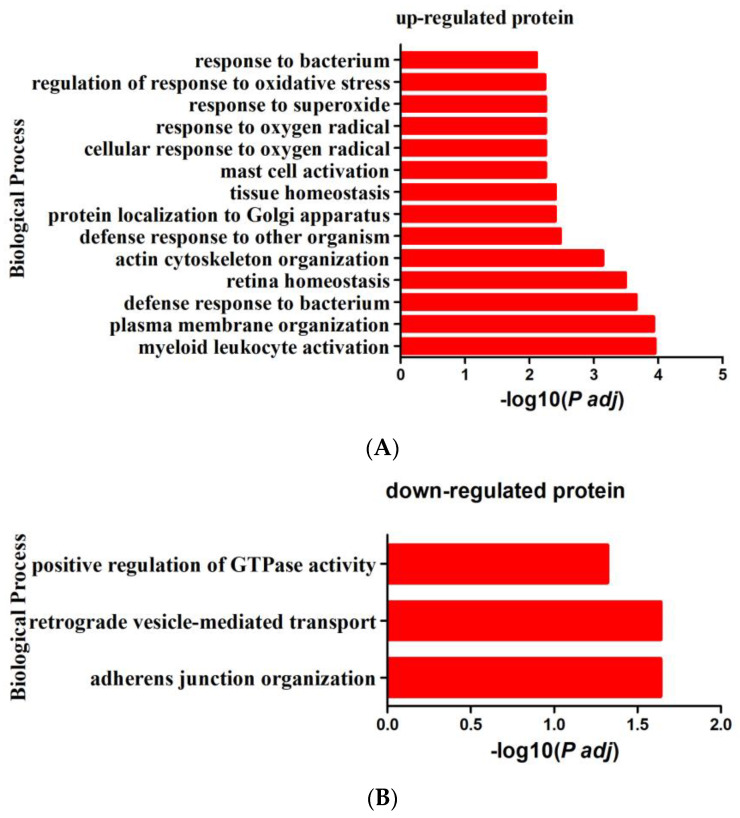
The results of functional enrichment analysis of DPPs. (**A**) Enriched GO-BP terms for up-regulated DPPs. (**B**) Enriched GO-BP terms for down-regulated DPPs. The x−axis is the −log10 *P* adj and the y−axis is the GO-BP terms in (**A**,**B**). (**C**) Enriched KEGG pathway for up-regulated DPPs. (**D**) Enriched KEGG pathway for down-regulated DPPs. The x−axis is the fold enrichment, and the y−axis is the KEGG pathways. The bubble size indicates the number of DPPs, and the color indicates the *p* value in (**C**,**D**).

**Figure 5 animals-13-00078-f005:**
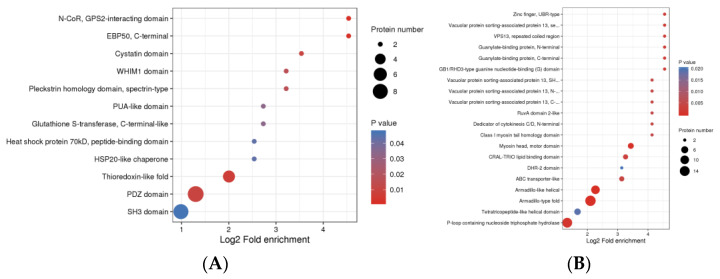
The top 25 protein domain enrichment analysis of DPPs. (**A**) The protein domain enrichment analysis of up-regulated DPPs. (**B**) The protein domain enrichment analysis of down-regulated DPPs. The x−axis is the Log2-fold enrichment, and the y−axis is the protein domains. The bubble size indicates the number of DPPs, and the color indicates the *p* value.

**Figure 6 animals-13-00078-f006:**
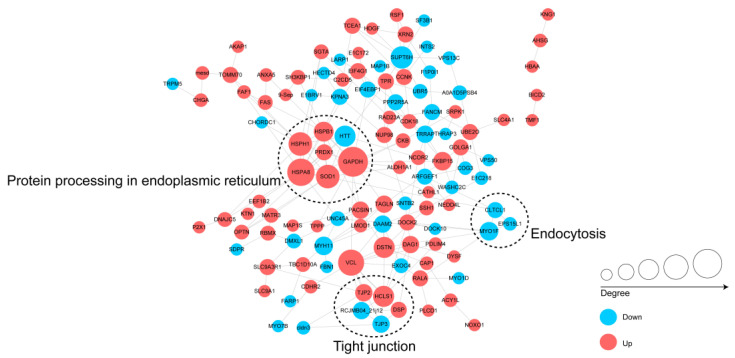
The PPIs networks of the identified DPPs. Red and blue indicate up- and down-regulated DPPs, respectively. The larger the circle, the more interaction nodes.

**Figure 7 animals-13-00078-f007:**
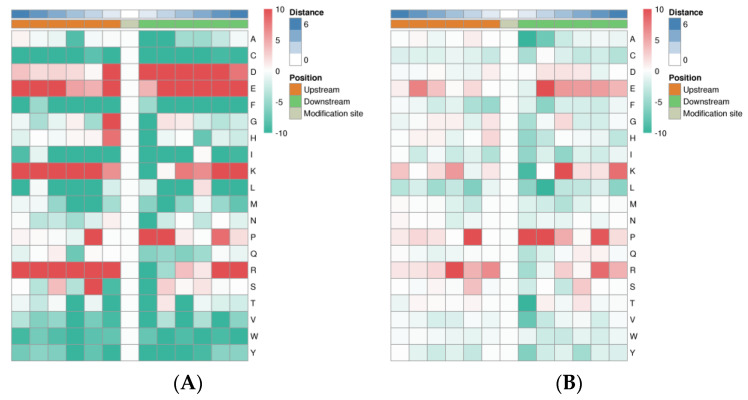
Motif enrichment heat maps of phospho-serine (**A**) and phospho-threonine (**B**) upstream and downstream of all identified phosphorylated modification sites. Red indicates significant enrichment of the amino acid near the modification site, while green indicates significant reduction of the amino acid near the modification site.

**Figure 8 animals-13-00078-f008:**
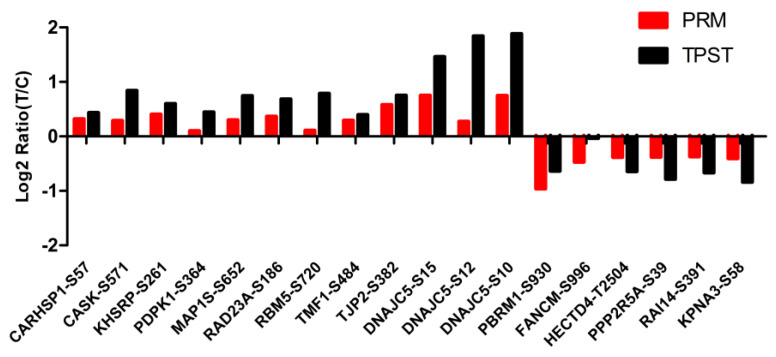
The validation results of the phosphorylation modification site with PRM. The x−axis is the significant phosphorylated modification site, and the y−axis is Log2-fold enrichment of treated /control (T/C) group.

## Data Availability

The raw data of phosphoproteomics profiles in this article are available in the NCGC Genome Sequence Archive with accession number OMIX001856 (https://ngdc.cncb.ac.cn/omix/preview/tnlv95Wp, accessed on 15 November 2022).

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
