# Peer review of "Phosphoproteomics Profile of Chicken Cecum in the Response to Salmonella enterica Serovar Enteritidis Inoculation"

_animals, 2022, doi:10.3390/ani13010078_

Round 1

Reviewer 1 Report

In this manuscript, Miao and Colleagues investigated This research combines TMT labeling, high performance liquid chromatography(HPLC)and mass spectrometry-based phosphoproteomics on cecum of the F1 cross of Guangxi Yao chicken (♀) and Jining Bairi chicken (♂). The results are interesting, but several minor points appear within the manuscript. The content of the manuscript is very novel and clear. But it needs to be replaced with a clearer picture.

Author Response

Point: The results are interesting, but several minor points appear within the manuscript. The content of the manuscript is very novel and clear. But it needs to be replaced with a clearer picture.

 Response: Thank you for your precious comments and advice. We are extremely grateful to you for your careful review. We have replaced more clearer pictures of Figure 2, 4, 7 in the Results chapter. And we carefully proof-read the manuscript to minimize typographical, grammatical, and bibliographical errors.

If there are any other modifications we could make, we would like very much to modify them and we really appreciate your help.

Reviewer 2 Report

Dear authors,

in my opinion the proposed manuscript is scientifically up-to-date, correct and clearly written. I have two suggestions to improve the proposed manuscript.
1. I suggest shortening the Abstract. It is too long.
2. I suggest removing Table 1 from the Results chapter. Table 1 can be found in the Appendix. Table 1 is too large for the manuscript.

Author Response

Point 1: I suggest shortening the Abstract. It is too long.

Response 1: We think this is an excellent suggestion. We have shortened the Abstract. The changes to the manuscript are given in the blue text.

Point 2: I suggest removing Table 1 from the Results chapter. Table 1 can be found in the Appendix. Table 1 is too large for the manuscript.

Response 2: We feel great thanks for your professional review work on our article. We agree with your assessment. We have removed Table 1 from the Results chapter.

Thank you again for your valuable and thoughtful comments. If there are any other modifications we could make, we would like very much to modify them and we really appreciate your help.

Reviewer 3 Report

The manuscript ” Phosphoproteomics profile of chicken cecum in the response to Salmonella enterica serovar Enteritidis inoculation” combines TMT labeling, high performance liquid chromatography(HPLC)and mass spectrometry-based phosphoproteomics to analyse the phosphoproteomics profile of chicken cecum in the response to Salmonella enterica serovar Enteritidis inoculation, Functional analysis and functional classification were performed for those differentially phosphorylated proteins (DPPs). The paper is generally well written and structured. However, in my opinion, the paper needs minor revisions as suggested below.

1.       L24: “Salmonella enterica, serovar Enteritidis” should be “Salmonella enterica serovar Enteritidis” ?

2.       Too little Introduction,please add some more on phosphoproteomics research.

3.       L76: the title “Animal inoculation and sample collection” shuold be capitalized.

4.       L78:How many chickens were used in this study? Please state the details.

5.       L85:Only three chickens per group were selected, is the sample number too small?

6.       L124: NH4OH should be NH4OH?

7.       L165: classified should be classify?

8.       L256: delete space before and after “-”?

9.       L357: add et al. after Huang.

10.   L360: add et al. after Shi.

11.   Line 447,465,469,477-480,486-489,497-507,518,520,532,537-540: title should  be small capital.

12.   Wrong format for the Abbreviated Journal Name.

Author Response

Point 1: L24: “Salmonella enterica, serovar Enteritidis” should be “Salmonella enterica serovar Enteritidis”?

Response 1: Thank you for pointing this out. The words“Salmonella enterica serovar Enteritidis” are right. We have corrected this mistake and the changes to the manuscript are given in the blue text. 

Point 2: Too little Introduction,please add some more on phosphoproteomics research.

Response 2: We think this is an excellent suggestion. We have added the researches of phosphoproteomics and the changes to the manuscript are given in the red text.

Point 3: L76: the title “Animal inoculation and sample collection” shuold be capitalized.

Response 3: Thanks for your careful checks. We are sorry for our carelessness. The title “Animal inoculation and sample collection” is capitalized now. And we checked all the titles carefully in this manuscript in order to ensure the titles are right.

Point 4: L78:How many chickens were used in this study? Please state the details.

Response 4: We apologize to you for being confused by our article. We used 90 chickens in this study, 38 chickens in the control group and 52 chickens in the treated group. We have added the details in the Materials and Methods chapter and the changes to the manuscript are given in the red text.

Point 5: L85:Only three chickens per group were selected, is the sample number too small?

Response 5: Thank you for your nice comments on our article. We chose 3 chickens for research because we referred to some articles about phosphoproteomics and proteomics of poultry and livestock. Three spleen tissues of duck were used to perform the phosphoproteomic analysis in the research of Tao et al. (Tao et al., 2020). And three samples were also used in the study of duck muscle proteomics (Zhang et al., 2021). Three liver samples were selected from each group in the study of liver phosphoproteomics (Zhao et al., 2021). The above researches are our partial references. And we will consider increasing the number of samples in the future experiments based on your suggestions.

References

Yun T, Hua J, Ye W, Ni Z, Chen L, Zhang C. The phosphoproteomic responses of duck (Cairna moschata) to classical/novel duck reovirus infections in the spleen tissue. Sci. Rep. 2020, 10(1):15315.

Zhang M, Wang D, Xu X, Xu W, Zhou G. iTRAQ-based proteomic analysis of duck muscle related to lipid oxidation. Poult. Sci. 2021, 100(4):101029.

Zhao XW, Zhu HL, Qi YX, Wu T, Huang DW, Ding HS, Chen S, Li M, Cheng GL, Zhao HL, Yang YX. Quantitative comparative phosphoproteomic analysis of the effects of colostrum and milk feeding on liver tissue of neonatal calves. J. Dairy Sci. 2021,104(7):8265-8275.

Point 6: L124: NH4OH should be NH4OH?

Response 6: Thank you for your careful work. We have replaced NH4OH with NH4OH.

Point 7: L165: classified should be classify?

Response 7: We are sorry for our negligence, and we have corrected the mistake. You can see the changes in the manuscript with blue text.

Point 8: L256: delete space before and after “-”?

Response 8: We have deleted space before and after “-”of “metabolism-other”.

Point 9: L357: add et al. after Huang.

Response 9: We agree with the your assessment. We have corrected the description and the changes to the manuscript are given in the blue text.

Point 10: L360: add et al. after Shi.

Response 10: We have changed description to make sentences more easy and smooth.

Point 11: Line 447,465,469,477-480,486-489,497-507,518,520,532,537-540: title should be small capital.

Response 11: Thank you again for your positive comments and valuable suggestions to improve the quality of our manuscript. The titles that you have referred were changed into small capital, except for the word at the beginning of title. We checked all the titles of the References chapter and corrected all the wrong format titles.

Point 12: Wrong format for the Abbreviated Journal Name.

Response 12: We are extremely grateful to you for your careful review. We have reconfirmed the format Abbreviated Journal Name and corrected the wrong format for all the Abbreviated Journal Name.

In addition, thank you again for your valuable and thoughtful comments. We have asked Dr Hai Lin, who's a well established expert, to polish our paper. We have carefully checked and improved the English writing in the revised manuscript. The revisions to the manuscript are given in the blue text, and the added contents are given in the red text. Please see whether the revised version met the English presentation standard.

If there are any other modifications we could make, we would like very much to modify them and we really appreciate your help.
